# Analysis of the Impact of Changes in Echo Signal Parameters on the Uncertainty of Distance Measurements in p-ToF Laser Rangefinders

**DOI:** 10.3390/s22165973

**Published:** 2022-08-10

**Authors:** Michał Muzal, Marek Zygmunt

**Affiliations:** Institute of Optoelectronics, Military University of Technology, 2 Sylwestra Kaliskiego Street, 00-908 Warsaw, Poland

**Keywords:** laser rangefinder, full-waveform LiDAR, precise distance measurements

## Abstract

The article presents results of research on the influence of changes in parameters of the digitally recorded echo signals on the uncertainty of pulsed Time-of-Flight (p-ToF) laser distance measurements. The main objective of the study was to evaluate the distance calculation method developed by the authors. This method is based on the acquisition of the full-waveform of the echo pulse signal and approximation of its shape by the second-degree polynomial (we called it SDPA for short). To determine the pulse transit time and measure the distance, the position of the vertex of this parabola is sought. This position represents the maximum intensity of the incoming echo signal and is related to the round-trip propagation time of the laser pulse. In the presented work, measurement uncertainty was evaluated using simulation tests for various parameters of the echo pulse. All obtained results were used to formulate the general relationship between the measurement uncertainty of the SDPA algorithm and the parameters of the received echo signals. This formula extends the base knowledge in the domain of laser p-ToF distance measurements. It can be used to estimate the measurement uncertainty of a FW LiDAR at an early design stage. This greatly improves capabilities of analysis of expected performance of the device. It can also be implemented directly into the rangefinder’s measurement algorithm to estimate the measurement uncertainty based on the emission of a single pulse rather than a series of pulses.

## 1. Introduction

Advances in digital signal acquisition and processing, particularly the development of high-speed analog-to-digital converters, have enabled their use in pulsed ToF laser rangefinders [1,2]. This technique is referred to as full-waveform acquisition of signals and is widely used in airborne and satellite LiDAR scanning of the Earth’s surface [3,4,5,6,7], and in self-driving vehicles [8]. Such use of analog-to-digital converters calls for efficient ways of extraction of echo signals and elimination of the discretization error introduced by sampling [9,10]. These methods depend on the sampling frequency and can result in the distance measurement error being as high as several meters, when low frequency sampling is used. In order to eliminate this error, it is important to recreate continuity of the time domain of received signals and derive its parametric formulation [11,12]. This can be undertaken through approximation of the shape of a laser pulse echo with a chosen fitting function. A similar approach was the basis for work presented in the article “Methods of Precise Distance Measurements for Laser Rangefinders with Digital Acquisition of Signals”, which led to the development of the “modified second−degree polynomial approximation” algorithm (SDPA-M) for real time processing of lidar waveforms [13]. The aim of the work presented in this article was to investigate the influence of changes in parameters of the received echo signal pulses on the precision of distance measurements using the SDPA algorithm. Since the algorithm is based on calculation of the position of the apex of a parabola fitted to the selected set of pulse samples, the parameters of the recorded pulse affect the uncertainty of distance measurements. We selected the three main parameters that characterize the digitally recorded signal, i.e., pulse full width at half maximum (FWHM), signal-to-noise ratio (SNR), and sampling frequency (*f_smp_*), and studied the influence of their changes on the measurement precision. Through our work we were able to formulate a general relationship linking the basic parameters of the echo pulse signal and receiver to the uncertainty of distance measurements with full-waveform LiDARs, in which (in order to improve measurement precision) an approximation of the sampled pulse with a second-degree polynomial is used.

## 2. Assumptions

The algorithm of the distance calculation evaluated in the following simulations was presented in detail in a previous article [13]. In this section, we briefly present the basic assumptions regarding this algorithm. The SDPA algorithm is based on the least square approximation of the shape of a digitally recorded echo of a light pulse by a second-degree polynomial. The reason why the second-degree polynomial was chosen, instead of more commonly used Gaussian function [14], is that it sufficiently models the shape of a laser pulse and is less demanding in terms of computational effort and, therefore, we assumed that it would result in a faster algorithm. The approximation is performed on the selected subset of *n* recorded samples that is expected to contain the returning echo of a probing pulse. In real-life applications this subset can be selected using one of the threshold discrimination methods [15,16,17,18]. Function fitting is performed using the least square method and returns coefficients of the quadratic function *y*(*x*) (parabola) that represents a mathematical model of the received pulse.
(1)$$y\left(x\right)=a_{2}x^{2}+a_{1}x+a_{0}$$
where: *y*(*x*) is the approximating polynomial function; *a*_0_, *a*_1_, *a*_2_ are its coefficients; and *x* ∈ *R* represents time.

Measurements of the distance to the target are based on determination of the *x_s_* time coordinate of the peak of the function *y*(*x*):(2)$$x_{s}=\frac{-a_{1}}{2a_{2}}$$

The *x_s_* coordinate is directly proportional to the round-trip time of the flight of a laser pulse to a target. The relationship between the calculated round-trip time of a pulse and the distance to the target is given by the formula:(3)$$s=\frac{c_{n}}{2}\cdot x_{s}$$
where: *s* [m] is the calculated distance; *x_s_* [s] is the time of occurrence of the maximum of approximated parabola; and *c**_n_* [m/s] is the speed of light in the chosen medium (air in the presented work).

A depiction of the basic principles of the SDPA is shown in Figure 1. The validity of using the SDPA algorithms in distance measurements in full-waveform lidars was proven in experiments described in a previous article [13]. The purpose of the work presented here was to examine the influence of changes in parameters of the received pulse signal on the measurement uncertainty $\sigma_{s}$ of the SDPA algorithm.

In the presented work, we assumed that all simulated pulses have a constant position (time of appearance) of the peak and that, in distance calculations, all available datapoints will always be used. In order to analyze the behavior of the SDPA algorithm, we disregarded all the effects that influence measurement uncertainty that are present in real-life applications, such as atmospheric turbulence and the shape of the target (topic discussed in [19]).

In real-life applications prior to the distance calculation, there is a need to locate and extract echo pulses from the recorded waveform. This can be particularly challenging in LiDAR forestry scanning applications and airborne and satellite altimetry where a single laser pulse can encounter multiple targets and yield overlapping echo waveforms [6,7]. In such cases, methods of waveform segmentation and data point extraction such as deconvolution [20] are needed, and advanced algorithms, such as Damped Gauss-Newton Multi-echo Decomposition, have been proposed [14]. We have assumed that, for single target measurements, simple threshold or constant-fraction discriminators are sufficient to extract datapoints of the echo signal.

After segmentation of the signal and selection of datapoints, a set of *n* samples is available to perform fitting of the parabola. In order to efficiently handle the set of samples, it is useful to write Equation (1) in a matrix form, where each row of matrices *X* and *Y* represents a single data point (*x_i_*, *y_i_*):(4)Y=X·A
where:$$X=\left[\begin{array}{c} \begin{array}{ccc} 1 & x_{1} & x_{1}{}^{2}\\ 1 & x_{2} & x_{2}{}^{2}\\ 1 & x_{3} & x_{3}{}^{2} \end{array}\\ \begin{array}{ccc} \cdots & \cdots & \cdots \end{array}\\ \begin{array}{ccc} 1 & x_{n} & x_{n}{}^{2} \end{array} \end{array}\right];\;A=\left[\begin{array}{c} a_{0}\\ a_{1}\\ a_{2} \end{array}\right];\;Y=\left[\begin{array}{c} y_{1}\\ y_{2}\\ \begin{array}{c} y_{3}\\ \begin{array}{c} \cdots \\ y_{n} \end{array} \end{array} \end{array}\right].$$

Parabola coefficients *a*_0_, *a*_1_, *a*_2_ are calculated by solving a regression problem by minimizing the least square error, that is, when all the partial derivatives of Equation (4) in respect to *a*_0_, *a*_1_, *a*_2_ equal 0. This requirement can be presented in a matrix form as:(5)$$\left(X^{T}X\right)\cdot A-X^{T}Y=0$$

The coefficients of the parabola are calculated using equation:(6)$$A={\left(X^{T}X\right)}^{-1}\cdot X^{T}Y$$

Statistical analysis of measurement uncertainty is typically performed by conducting a series of trials with mean and standard deviation calculations. In the presented work, we aimed to examine the effect of changes in parameters of the pulse on distance measurements; therefore, in a typical approach, to gain a statistically significant set of measurements, we would need to perform a series of trials for each set of chosen parameters of the pulse. However, it is possible to estimate the measurement uncertainty from a single simulation, which greatly increases the speed. To do so, the standard deviations $\sigma_{a_{1}}$ and $\sigma_{a_{2}}$ of the coefficients *a*_1_ and *a*_2_ can be calculated using formulas:(7)$$\sigma_{a1}=\sqrt{Var\left(a_{1}\right)}=\sigma_{y}\sqrt{\frac{n\sum_{i=1}^{n}x_{i}{}^{4}-{\left(\sum_{i=1}^{n}x_{i}{}^{2}\right)}^{2}}{Det\left(X^{T}X\right)}}$$
(8)$$\sigma_{a2}=\sqrt{Var\left(a_{2}\right)}=\sigma_{y}\sqrt{\frac{n\sum_{i=1}^{n}x_{i}{}^{2}-{\left(\sum_{i=1}^{n}x_{i}\right)}^{2}}{Det\left(X^{T}X\right)}}$$
where:

$\sigma_{y}$ is the standard deviation of signal’s noise,
$$Det\left(X^{T}X\right)=n\overset{n}{\underset{i=1}{\sum }}x_{i}{}^{2}\overset{n}{\underset{i=1}{\sum }}x_{i}{}^{4}+2\overset{n}{\underset{i=1}{\sum }}x_{i}\overset{n}{\underset{i=1}{\sum }}x_{i}{}^{2}\overset{n}{\underset{i=1}{\sum }}x_{i}{}^{3}-{\left(\overset{n}{\underset{i=1}{\sum }}x_{i}{}^{2}\right)}^{3}-n{\left(\overset{n}{\underset{i=1}{\sum }}x_{i}{}^{3}\right)}^{2}-{\left(\overset{n}{\underset{i=1}{\sum }}x_{i}\right)}^{2}\overset{n}{\underset{i=1}{\sum }}x_{i}{}^{4}$$

Finally, the uncertainty of distance measurements can be calculated using:(9)$$\sigma_{s}=\frac{c_{\mu }}{2}\cdot \sqrt{{\left(\frac{-1}{2a_{2}}\sigma_{a_{1}}\right)}^{2}+{\left(\frac{a_{1}}{2{\left(a_{2}\right)}^{2}}\sigma_{a_{2}}\right)}^{2}}$$

By using Equation (9) to estimate uncertainty during the presented simulations, we have significantly reduced the number of trials needed to correctly assess the impact of changes in the recorded echo signal on the final measurement uncertainty.

## 3. Methodology of Experiments

There are three main parameters of the digitally recorded echo signals that describe the pulse signal and the electro-optical receiver and affect the precision of p-ToF distance measurements. These parameters are the signal-to-noise ratio, the duration of the probing pulse measured as its full width at half maximum, and the sampling frequency of the used analog-to-digital converter [21]. The signal-to-noise ratio in a laser rangefinder is affected by the peak power of the returning echo signal (influenced by the distance to the target, its reflectivity and shape, and atmospheric conditions) and the ability of a receiver to collect as much energy of the returning signal as possible with the lowest possible self-introduced noise [18,19]. Echo pulse width, in combination with sampling frequency, affects the number of signal samples available for use by the SDPA algorithm. In the presented work, the effect of changes in those parameters on the uncertainty of measurements using the SDPA algorithm was investigated in simulation experiments. We calculated the $\sigma_{s}$ uncertainty as a function of:Signal-to-noise ratio—$\sigma_{s}$(SNR), where SNR $=A/\sigma_{y}$,Sampling frequency—$\sigma_{s}\left(f_{smp}\right)$,Echo pulse FWHM—$\sigma_{s}\left(\tau \right)$.

Values of samples of the simulated echo pulse were calculated using a function:(10)$$f\left(x_{i}\right)=A\cdot \cos^{2}\left(\frac{\pi \;}{2\cdot \tau }\cdot x_{i}\right)+\xi \left(x_{i}\right)$$
where: $x_{i}=j\cdot T$ is a discrete time of acquisition of the *i*th sample, $i\in Z^{0+},$
*T* is the sampling period, j=(i−(n/2)) and n=⌊2·τ/T⌋ is the required number of samples of generated signal; ⌊ ⌋ is the round down (floor) operator; *A* is the peak value of the generated pulse; and $\xi \left(x_{i}\right)$ is a random, normally distributed noise *N*(0, *σ_y_*^2^) with mean value 0 and variance *σ_y_*^2^.

Depending on the relationship between the required width of the pulse and sampling frequency, the *n* number of signal samples can be even or odd. In both cases, a symmetrical distribution of samples is maintained around time zero, which is the expected position of the peak of simulated pulse. Figure 2 presents an example of two sets of samples generated when *n* is an even (*n* = 10) and odd (*n* = 25) number. It is worth noting that in both cases the actual peak of the signal remains at *x_s_* = 0 [s]. By changing input parameters in Formula (10), we directly change the width of the generated pulse and the sampling frequency. The SNR can be varied either by changing the peak value *A* of a signal or the standard deviation *σ_y_* of the noise. The minimal number of samples needed for the SDPA algorithm is *n* = 3.

Simulation experiments were carried out on the personal computer using the MATLAB tool. The algorithm of the simulation process is shown in Figure 3. Simulations were started with a set of initial parameters used to generate a set of signal samples with the desired signal-to-noise ratio. That set was used as input data for the SDPA algorithm and calculation of the distance and predictions of its uncertainty was performed. Three blocks of trials were conducted. In each block, the influence of changes in a single, chosen parameter on the measurement uncertainty was examined in several trials. For each trial, the remaining two parameters were set as constant. In the following sections, results of each block of trials are presented. Figure 3 illustrates the procedure taken in each trial within a single block of measurements.

## 4. Results

In this section, the results of each block of trials are presented as plots of functions $\sigma_{s}\left(\tau \right)$, $\sigma_{s}\left(f_{smp}\right)$ and $\sigma_{s}\left(SNR\right)$.

### 4.1. Examination of the Influence of Changes in Signal-to-Noise Ratio on the Measurement Uncertainty $\sigma_{s}$(SNR)

In this set of trials, the influence of changes in the signal-to-noise ratio on the uncertainty of distance measurements was examined. Two blocks of trials were performed, each consisting of four series of simulation runs. In block 1, during each series of simulations the sampling frequency was fixed, and pulse width was set to four different values, one for each series. In block 2, we fixed the pulse width and changed the sampling frequency for each series. In both cases, the SNR was changed continuously between 10 and 100 with 0.1 resolution. Table 1 describes parameters of the test signals generated during trials.

The results of the conducted simulations are shown in Figure 4 and Figure 5. They show characteristics of the relationship between measurement uncertainty of the SDPA algorithm and the signal-to-noise ratio (SNR). As expected, the uncertainty decreases with increasing SNR and the relationship is non-linear. We evaluated that the uncertainty of measurements of the distance with SDPA algorithm is inversely proportional to the signal-to-noise ratio. This dependency can be written as:(11)$$\sigma_{s}\sim \frac{1}{SNR}\;$$

Plots in Figure 4 and Figure 5 show that the uncertainty $\sigma_{s}$ also depends on the sampling frequency *f_smp_* and width *τ* of a pulse. Precision of measurements increases with increasing sampling frequency and narrowing of the pulse. In the following sections, we determine the characteristics of these changes.

### 4.2. Examination of the Influence of Changes in Sampling Frequency f_smp_ on the Measurement Uncertainty $\sigma_{s}\left(f_{smp}\right)$

In this block of trials, the influence of changes in the sampling frequency *f_smp_* on the uncertainty of distance measurements was examined. Two blocks of trials were performed, each consisting of four series of simulation runs. In block 3, during each series of simulations, the signal-to-noise ratio was fixed, and the pulse width was set to four different values, one for each series. In block 4, we fixed the pulse width and for each series set a different SNR. In both cases the sampling frequency was changed continuously between 100 and 1000 MHz with 1 Hz resolution. Table 2 describes parameters of the test signals generated during the trials.

The results of the conducted simulations are shown in Figure 6 and Figure 7. They show characteristics of the relationship between measurement uncertainty of the SDPA algorithm and sampling frequency (*f_smp_*). Uncertainty decreases with increasing sampling frequency (decreasing sampling period). The relationship $\sigma_{s}\left(f_{smp}\right)$ is non-linear. We evaluated that the uncertainty of measurements of the distance with SDPA algorithm is directly proportional to the square root of the sampling period or reciprocal of the sampling frequency. These dependencies can be written as:(12)$$\sigma_{s}\sim \sqrt{\frac{1}{f_{smp}}}\;\mathrm{and}\;\sigma_{s}\sim \sqrt{T}$$

Similar to the results of simulations shown in Section 4.1, the plots in Figure 6 and Figure 7 also show that the uncertainty $\sigma_{s}$ depends on the width *τ* of a pulse and confirm the SNR dependency of relationship (11). In the following section, the dependency of the uncertainty of measurements from the pulse width (FWHM) are examined.

### 4.3. Examination of the Influence of Changes in Pulse Width on the Measurement Uncertainty $\sigma_{s}\left(\tau \right)$

In this block of trials, the influence of changes in the pulse width (measured as full width at half maximum (FWHM)) on the uncertainty of distance measurements using the SDPA algorithm was examined. Two blocks of trials where performed, each consisting of four series of simulation runs. In block 5, during each series of simulations the signal-to-noise ratio was fixed, and the sampling frequency was set to four different values, one for each series. In block 6, we locked the sampling frequency and set a different SNR for each series. In both cases the width of generated pulses was changed continuously between 5 ns and 100 ns with 0.1 ns resolution. Table 3 describes parameters of the test signals generated during the trials.

The results of conducted simulations are shown in Figure 8 and Figure 9. They show that the relationship $\sigma_{s}\left(\tau \right)$ between measurement uncertainty of the SDPA algorithm and width *t* of generated pulses is non-linear. Uncertainty increases with increasing FWHM of a pulse. We evaluated that the uncertainty of measurements is directly proportional to the square root of the width of a pulse. This dependency can be written as:(13)$$\sigma_{s}\sim \sqrt{\tau }\;$$

Results of simulations also confirm the relationships of Equations (11) and (12), as discussed in Section 4.1 and Section 4.2.

All of the presented results show basic relations between parameters of the echo signal and distance measurements performed using the SDPA algorithm. In the following section, the synthesis of Equations (11)–(13) that leads to the general formula for measurement uncertainty $\sigma_{s}\left(\tau ,f_{smp},SNR\right)$ is presented.

## 5. Discussion

Based on the simulation results, we determined the relationships between the uncertainty of distance measurements using the SDPA algorithm and the parameters of echo signals and acquisition system. Combining the relations of Equations (11)–(13) leads to the following relation:(14)$$\sigma_{s}\sim \frac{1}{SNR}\cdot \sqrt{\frac{\tau }{f_{smp}}}\;\mathrm{or}\;\sigma_{s}\sim \frac{1}{SNR}\cdot \sqrt{\tau \cdot T}$$

It follows from the above relationships that if, in a digital signal processing system, the uncertainty of determining the sample acquisition time is negligibly small relative to the uncertainty of determining its value (the clock jitter of the ADC is negligibly small), then the uncertainty of determining the distance using the SDPA algorithm is directly proportional to the root of the quotient of the pulse width and sampling frequency, and inversely proportional to the signal-to-noise ratio. The relationship of Equation (14) is directly related to the measurement of the time-of-flight and its unit is the second. Analogous to Equation (3), we relate the uncertainty of the time measurement from the relation of Equation (14) to the uncertainty of the distance measurement by multiplying it by the value of the speed of light in the selected media. This leads to the relationship:(15)$$\sigma_{s}\sim \frac{c_{\mu }}{2}\left(\frac{1}{SNR}\cdot \sqrt{\frac{\tau }{f_{smp}}}\right)$$

In order to transform the relationship of Equation (15) into an equation that allows for prediction of distance measurement uncertainty of the SDPA algorithm, we introduced the coefficient *k*:(16)$$\sigma_{s}=\frac{k\cdot c_{\mu }}{2\cdot SNR}\cdot \sqrt{\frac{\tau }{f_{smp}}}$$

The value of the coefficient *k* was estimated by fitting (using the least squares method) the curves determined by Equation (16) to the data collected during simulation. In the case of the presented results, the value of *k* was equal to 0.536. We determined that the uncertainty of the SDPA algorithm also depends on the number of samples extracted from the waveform by a threshold selection. The most commonly used method—the constant fraction discrimination—sets the threshold value at 50% of the peak value of the pulse, resulting in the 50% point extraction ratio. We determined that, in this case, the value of the *k*-factor must be increased and should be equal to *k* = 1.0.

In Figure 10, Figure 11 and Figure 12, plots of changes in the uncertainty of distance measurements are presented. Each plot is a result of multiple calculations of standard deviation $\sigma_{s}$ using Equation (16) with various input parameters that are described in Table 4.

The comparison of the graphs from Figure 10, Figure 11 and Figure 12 with the graphs presenting the simulation data (Figure 4, Figure 5, Figure 6, Figure 7, Figure 8 and Figure 9) shows that estimations performed using Equation (16) are sufficiently accurate and reassemble the behavior of the SDPA algorithm. The derived relationship of Equation (15) and Equation (16) represent the novel contribution to the field. Using Equation (16), it is possible to estimate the precision of distance measurements using a full-waveform LiDAR, with the echo pulse signal expressed as a second-degree polynomial, based only on the parameters of the echo pulse and the receiver. In order to verify whether Equation (16) can be applied to real-life measurements, we attempted to predict the uncertainty of measurements presented in [13,14] using Formula (16). In the work presented in the article [13], in order to evaluate the precision of the SDPA algorithm, we have performed five trials with different sets of parameters of generated pulses and different SNR ratios. We used the constant fraction discrimination method to extract pulse datapoints from the waveform. Each trial consisted of *L* = 1000 simulated measurements of distance. In Table 5 we present results of those simulation trials and results of standard deviation estimations undertaken with the help of Equation (16). The value of *k* = 1.0 was chosen because of the 50% point extraction ratio caused by the constant fraction discrimination method. Both values of the standard deviation that were determined experimentally and analytically are close to each other in all five scenarios, confirming the effectiveness of the proposed formula.

Equation (16) can also be used to compare the precision of other algorithms used in FW-LiDARs with the expected precision of the SDPA algorithm. In a recently published article [14], the authors present a full waveform LiDAR system and propose the Damped Gauss–Newton Decomposition algorithm to efficiently extract datapoints from overlapping waveforms and measure distance in a muti-target environment and low SNR conditions. To evaluate the proposed Equation (16), we chose one result among those presented in the article and attempted to predict and compare the uncertainty of the SDPA algorithm under the same signal conditions. The authors of [14] showed that the standard deviation of the distance measurement achieved by their FW-LiDAR system was equal to $\sigma_{s}=6.8\;\mathrm{cm}$ under the following conditions:
-no overlapping of the echo signals, 3 m spacing between targets;-SNR = 20 dB (which in our case represents SNR = 10);-sampling frequency $f_{smp}=1\;\mathrm{GSa}/s$;-pulse width at half maximum τ=10 ns.

By substituting these values into Equation (16), we obtain the value of the expected measurement precision of the SDPA algorithm used in the same system. Assuming the use of the constant fraction discrimination method, we set the value of *k* = 1.0. All this resulted in the expected standard deviation of measurements $\sigma_{s}=4.7\;\mathrm{cm}$. This value can be viewed as the best achievable standard deviation of measurements in the situation when no additional sources of measurement error are present, beside normally distributed noise. We also assume that the system corrects for any mean (bias) errors.

The ability to predict the expected standard uncertainty of a device can be helpful in assessing the magnitude of the influence of other sources of uncertainty on the total measurement error. When properly calibrated with the designed LiDAR system, Equation (16) can be used to estimate the uncertainty of distance measurements performed by the emission of a single laser pulse. This feature may be desirable in all systems where there is no possibility of multiple measurements.

## 6. Conclusions

In the presented work, the influence of changes in parameters of pulse echo signals on the measurement precision of the SDPA algorithm was analyzed via computer simulations. Simulations were performed to examine the influence of the three most important parameters of the echo signal, i.e., signal-to-noise ratio, sampling frequency, and pulse width. A general equation (Equation (16)) that binds measurement precision with parameters of the echo pulse signal was proposed. This formula allows the estimation of the precision of measurements of the FW-LiDAR at an early design stage, such as during the selection of components. It clearly shows that achieving high measurement precision requires an effort to design systems with high signal and low-noise, using narrow laser pulses and ADCs with a high sampling rate. All of this can be a challenging task but, using the presented formula, the system designer is able to make the right choices when shaping the design requirements. In the authors’ opinion, this is a unique feature that extends the base knowledge in the domain of p-ToF measurements of distance. Authors have benefited from the presented equation and the SPDA algorithm in the works described in articles [22,23,24,25].

## Figures and Tables

**Figure 1 sensors-22-05973-f001:**
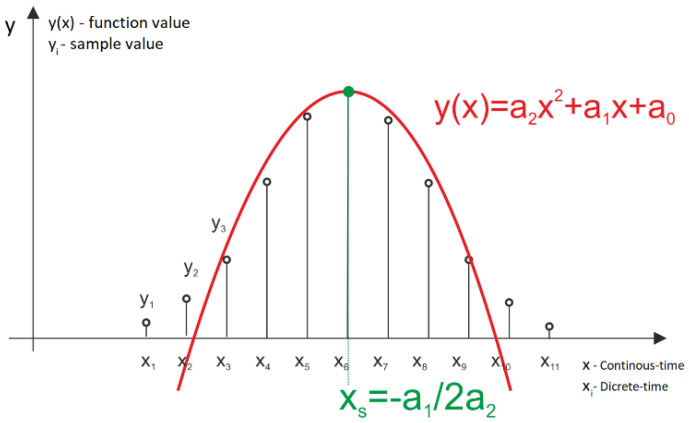
Approximation of a sampled signal with a second−degree polynomial (SDPA).

**Figure 2 sensors-22-05973-f002:**
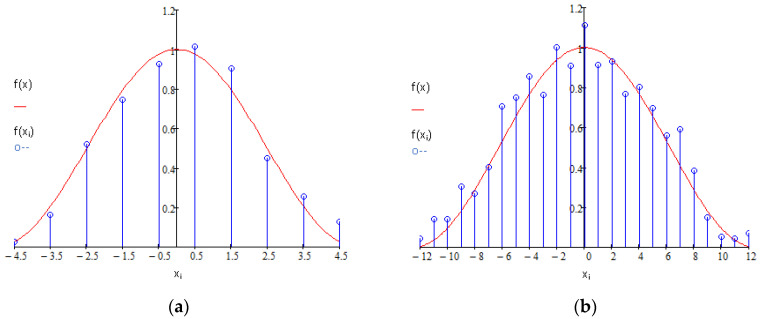
Example of the pulse signal functions *f*(*x*) and sample sets *f*(*x_i_*) used in the simulations. The plot was drawn for (**a**) even (*A* = 1, *τ* = 50 ns, *f_s_* = 50 MHz, noise $\xi \left(x_{i}\right)\in N\left(0,0.1)\right)$ and (**b**) odd (*A* = 1, *τ* = 50 ns, *f_s_* = 250 MHz, noise $\xi \left(x_{i}\right)\in N\left(0,0.1)\right)$ numbers of samples.

**Figure 3 sensors-22-05973-f003:**
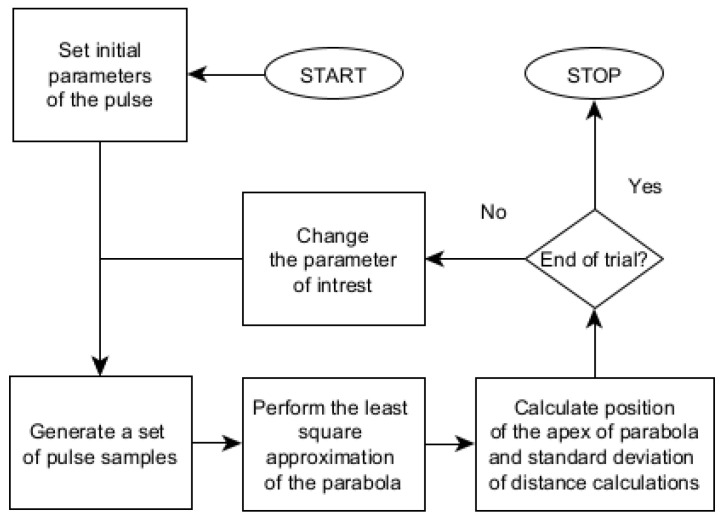
Algorithm of simulation trials.

**Figure 4 sensors-22-05973-f004:**
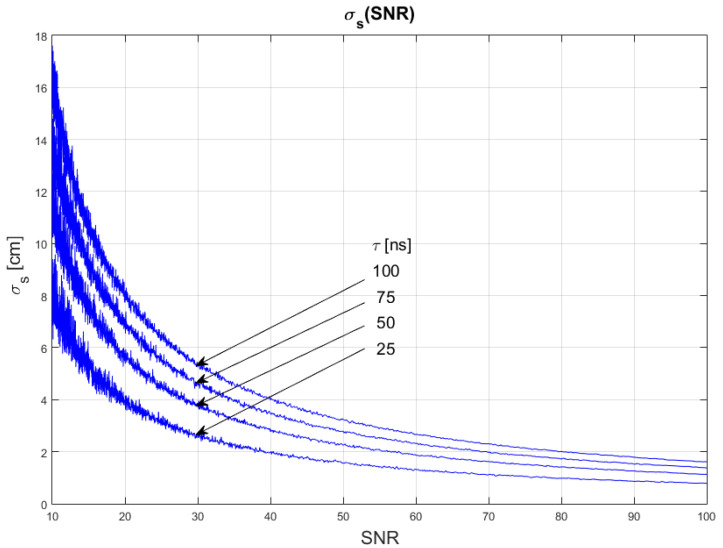
Dependence of measurement uncertainty on the signal-to-noise ratio (SNR). Parameters of generated pulses are shown in Table 1 Block 1.

**Figure 5 sensors-22-05973-f005:**
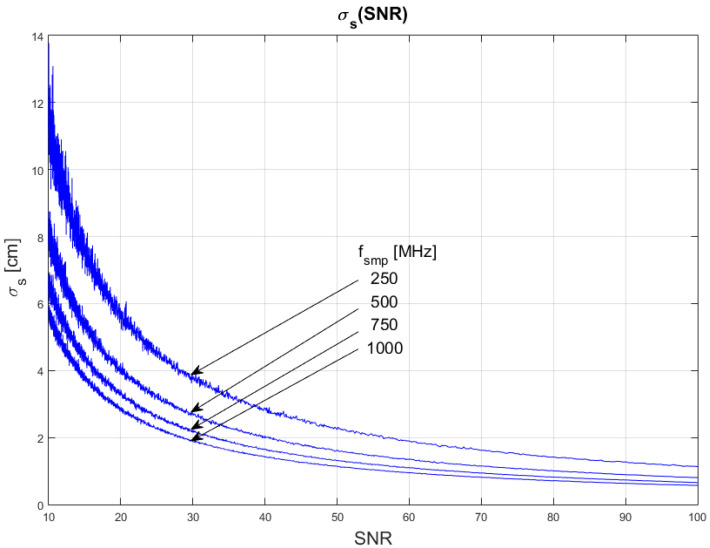
Dependence of measurement uncertainty on the signal-to-noise ratio (SNR). Parameters of generated pulses are shown in Table 1 Block 2.

**Figure 6 sensors-22-05973-f006:**
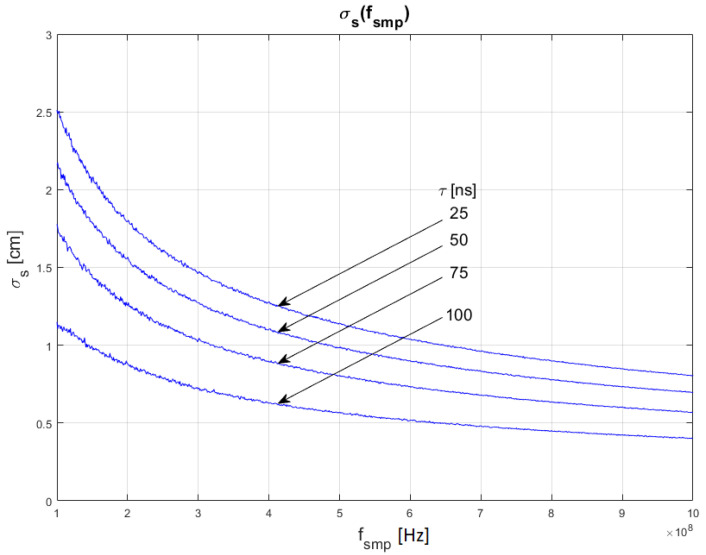
Dependence of measurement uncertainty on sampling frequency *f_smp_*. Parameters of generated pulses are shown in Table 2 Block 3.

**Figure 7 sensors-22-05973-f007:**
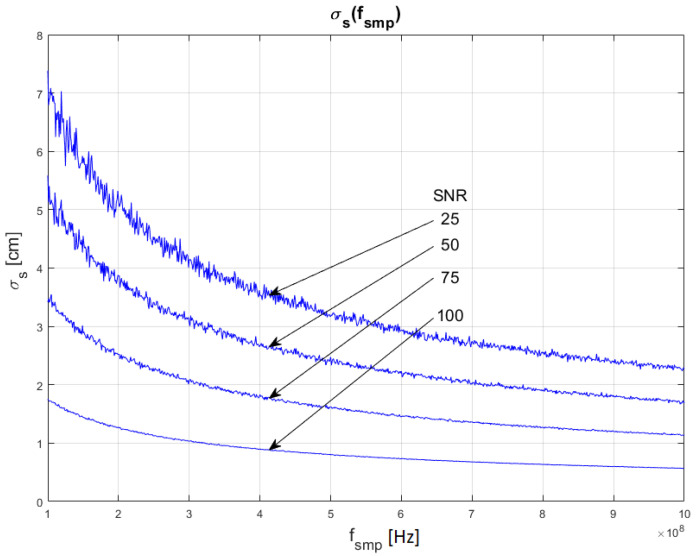
Dependence of measurement uncertainty on sampling frequency *f_smp_*. Parameters of generated pulses are shown in Table 2 Block 4.

**Figure 8 sensors-22-05973-f008:**
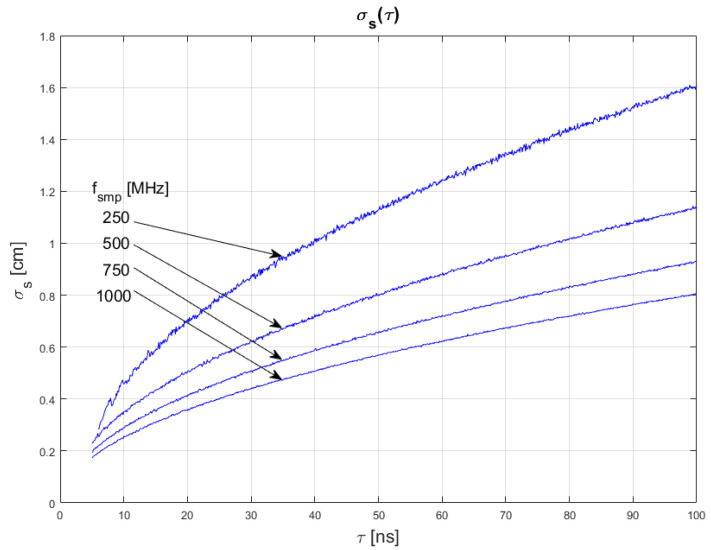
Dependence of measurement uncertainty on pulse width for *τ* for SNR = 100. Parameters of generated pulses are shown in Table 3 Block 5.

**Figure 9 sensors-22-05973-f009:**
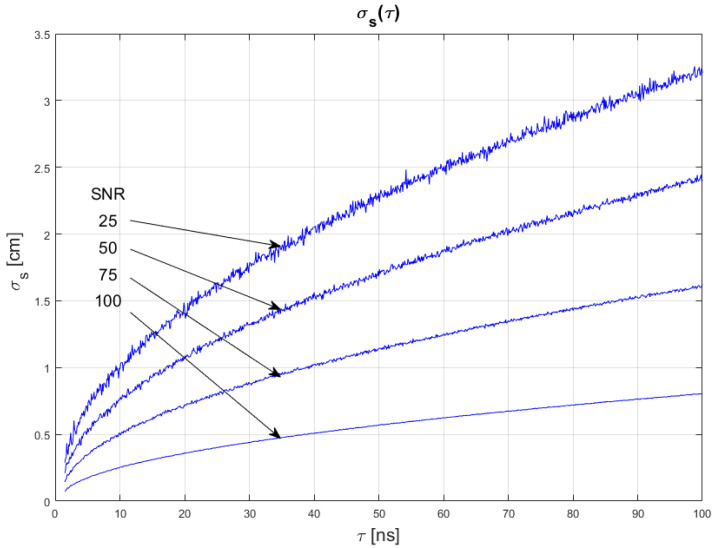
Dependence of measurement uncertainty on pulse width *τ* for *f_smp_* = 1 GHz. Parameters of generated pulses are shown in Table 3 Block 6.

**Figure 10 sensors-22-05973-f010:**
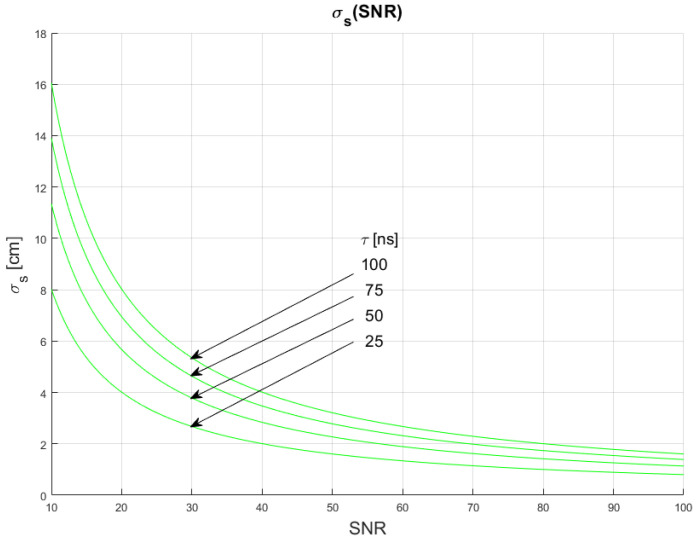
Plots of the dependance of the measurement’s uncertainty calculated using Equation (16) for signal parameters shown in Table 4. Comparable results are shown in Figure 4.

**Figure 11 sensors-22-05973-f011:**
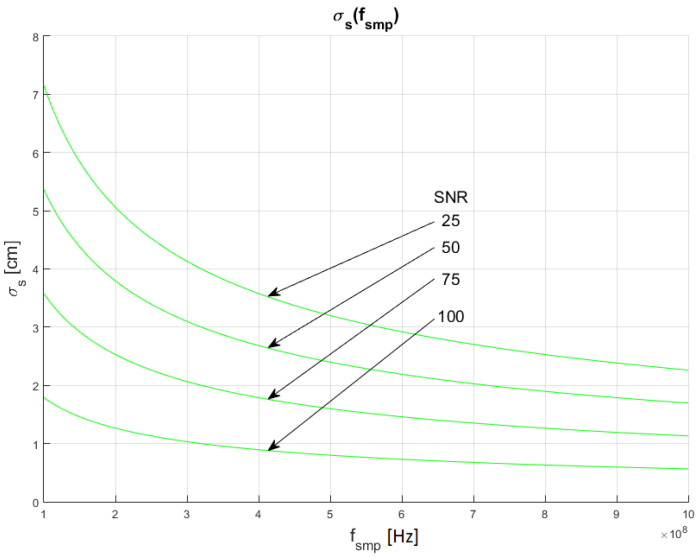
Plots of the dependance of the measurement’s uncertainty calculated using Equation (16) for signal parameters shown in Table 4. Comparable results are shown in Figure 7.

**Figure 12 sensors-22-05973-f012:**
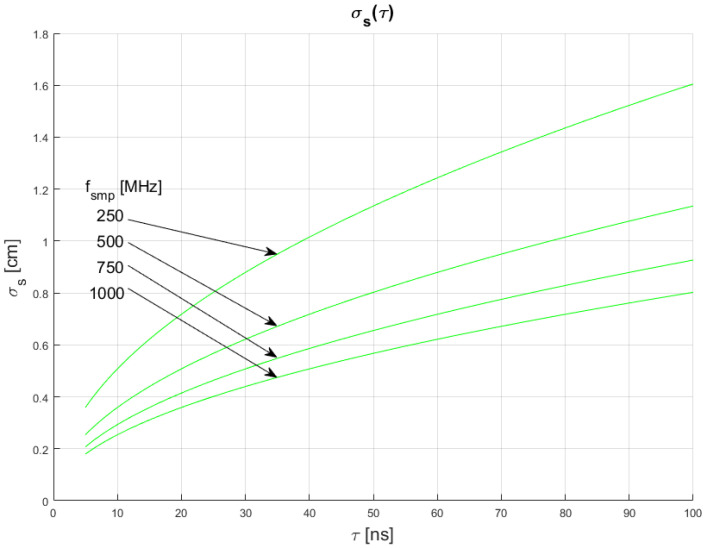
Plots of the dependance of the measurement’s uncertainty calculated using Equation (16) for signal parameters shown in Table 4. Comparable results are shown in Figure 8.

**Table 1 sensors-22-05973-t001:** Parameters of pulses generated during trials.

Block Number	SNR	τ [ns]	$f_{smp}$ [MHz]
1	10–100	25, 50, 75, 100	250
2	10–100	50	250, 500, 750, 1000

**Table 2 sensors-22-05973-t002:** Parameters of pulses generated during trials.

Block Number	SNR	τ [ns]	$f_{smp}$ [MHz]
3	100	25, 50, 75, 100	100–1000
4	25, 50, 75, 100	50	100–1000

**Table 3 sensors-22-05973-t003:** Parameters of pulses generated during trials.

Block Number	SNR	τ [ns]	$f_{smp}$ [MHz]
5	100	5–100	250, 500, 750, 1000
6	25, 50, 75, 100	5–100	1000

**Table 4 sensors-22-05973-t004:** Parameters of signals assumed during calculations using Equation (16) for *k* = 0.536.

Figure Number	SNR	τ [ns]	$f_{smp}$ [MHz]
10	10–100	25, 50, 75, 100	250
11	25, 50, 75, 100	50	100–1000
12	100	5–100	250, 500, 750, 1000

**Table 5 sensors-22-05973-t005:** Comparison of the results of simulation trials presented in article [13] with uncertainty estimated using Equation (16).

Trial Number	1	2	3	4	5
Sampling frequency *f_pr_* (MHZ)	333				
FWHM *τ* [ns]	39	78	156	39	156
SNR	181			631	
Standard deviation of the trial-data analysis, *σ_s_* (cm)	1.05	1.17	1.70	0.27	0.46
Calculated standard deviation *σ_s_* (cm), *k* = 1.0	0.89	1.27	1.79	0.26	0.51

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
