# Peer review of "Analysis of the Impact of Changes in Echo Signal Parameters on the Uncertainty of Distance Measurements in p-ToF Laser Rangefinders"

_sensors, 2022, doi:10.3390/s22165973_

Round 1
Reviewer 1 Report
Before closing the introduction, the authors should include a list of contributions to this research.
The cdot in the Eq. (3), Does it have an especially meaning?
The axes in Figure 1, Do it represent discrete values on both axes?
Equation(5) should be explained in terms of a regression problem solved by minimizing the Least Square Error, valid for N>>3.
Line 105 should be corrected. Please, make a deep proofread on entire document.
Please, include more details about deriving Eqs. (6) to (8), and prove the importance of (8).
Tha authors must make a double-check of equations, including the related operative conditions.
In Table 1, a simbol around 50 in column 3 and line 3 should be referenced and explained.
Labes should be corrected, see "Tab.1 describes parameters..." must be "Table 1 describes parameters". Figure.6 and 214 Figure.7 should be Figure 6 and Figure 7. The authors should consider to use latex command \Cref to automitize the labeling and references.
The authors use referencing article [9], many details must be included in those study to support and justify the originality of this work.
A subsection to include the discussions must be added to the manuscript.
In general, the mathematical formalizaton must be substantially improved.
Many bibliographical references should be updated, considering the relevant contribution from last five years related to this study.
Reviewer 2 Report
This paper investigates the SDPA distance calculation method for laser range finder by changing different parameters of the received echo signals pulse. This method is based on the acquisition of the full waveform of the echo pulse signal and approximation of its shape by the second-degree polynomial.
Comments:
1- Line 12: what SDPA refers to? Please state what SDPA stands for.
2- In the introduction section: it is preferred to mention more state of the art related to the area of this study.
3- Line 77: Cm is not found in equation 3.
4- Equation 3: There may be unbalance of the units in this equation; What is the unit of Xs, is it in meters or has no units?
5- Line 95: please remove the space after “Figure 1” and make “and 2” in the same line.
6- Line 144 to line 146: Please justify why the author selects these values for the nsig.
7- Line 274: will the k coefficient be fixed with the value 0.536, or it is just for the used data here in this article?
8- Could this be applied to real data instead of using simulated data?
Round 2
Reviewer 1 Report
The authors made a good effort, addressing correctly most of my concerns. The actual version of the manuscript is technically sound.
Author Response
Thank You for your efforts in improving the manuscript